# Beyond shallow feelings of complex affect: Non-motor correlates of subjective emotional experience in Parkinson's disease

**Claudia Carricarte Naranjo**[1,2]*, **Claudia Sánchez Luaces**[1], **Ivonne Pedroso Ibáñez**[3], **Andrés Machado**[1], **Hichem Sahli**[2,4], **María Antonieta Bobes**[5]

**1** Facultad de Biología, Universidad de La Habana, La Habana, Cuba, **2** Department of Electronics and Informatics, Vrije Universiteit Brussel, Brussel, Belgium, **3** Centro Internacional de Restauración Neurológica (CIREN), La Habana, Cuba, **4** Interuniversity Microelectronics Centre (IMEC), Heverlee, Belgium, **5** Centro de Neurociencias de Cuba, La Habana, Cuba

* carri@fbio.uh.cu

**Data Availability Statement:** The DOI necessary to access the data related to this manuscript is: 10.5281/zenodo.7630731.

## Abstract

Affective disorders in Parkinson's disease (PD) concern several components of emotion. However, research on subjective feeling in PD is scarce and has produced overall varying results. Therefore, in this study, we aimed to evaluate the subjective emotional experience and its relationship with autonomic symptoms and other non-motor features in PD patients. We used a battery of film excerpts to elicit Amusement, Anger, Disgust, Fear, Sadness, Tenderness, and Neutral State, in 28 PD patients and 17 healthy controls. Self-report scores of emotion category, intensity, and valence were analyzed. In the PD group, we explored the association between emotional self-reported scores and clinical scales assessing autonomic dysregulation, depression, REM sleep behavior disorder, and cognitive impairment. Patient clustering was assessed by considering relevant associations. Tenderness occurrence and intensity of Tenderness and Amusement were reduced in the PD patients. Tenderness occurrence was mainly associated with the overall cognitive status and the prevalence of gastrointestinal symptoms. In contrast, the intensity and valence reported for the experience of Amusement correlated with the prevalence of urinary symptoms. We identified five patient clusters, which differed significantly in their profile of non-motor symptoms and subjective feeling. Our findings further suggest the possible existence of a PD phenotype with more significant changes in subjective emotional experience. We concluded that the subjective experience of complex emotions is impaired in PD. Non-motor feature grouping suggests the existence of disease phenotypes profiled according to specific deficits in subjective emotional experience, with potential clinical implications for the adoption of precision medicine in PD. Further research on larger sample sizes, combining subjective and physiological measures of emotion with additional clinical features, is needed to extend our findings.

**Funding:** The study was supported by a Vlaamse Interuniversitaire Raad voor Universitaire Ontwikkelingssamenwerking (VLIR-UOS) TEAM project between the Cuban Center for Neurosciences (Cuba), the Vrije Universiteit Brussel (VUB) (Belgium), and the Ghent University (Belgium), and a Doctoral Exchange Grant to CCN by the VUB International Relations and Mobility Office (VUB-IRMO) (Belgium). The funders had no role in study design, data collection and analysis, decision to publish, or preparation of the manuscript.

**Competing interests:** The authors have declared that no competing interests exist.

## Introduction

Parkinson's disease (PD) is a progressive neurodegenerative disorder characterized by the presence of motor symptoms that result from the selective loss of dopaminergic neurons in the substantia nigra pars compacta. The nigral damage, however, is accompanied by extensive extranigral pathology that involves multiple neurotransmitter pathways within the brain and peripheral autonomic nervous system, leading to non-motor symptoms which are a critical component of the disease and often precede the motor changes [1,2]. Among the non-motor abnormalities, cognitive impairment, neuropsychiatric disorders, sleep dysfunction, and autonomic dysregulation have a significant role in the disease process.

Emotional disturbances are also part of the non-motor symptom complex of PD and concern all components of emotion [3]. Deficits in recognizing emotions from both the face and the voice, besides altered production of emotional facial expressions and speech, have been extensively documented in PD [4–8]. Likewise, impaired physiological responses to emotion have also been reported [9–12]. Considering different levels of affective processing, a previous study revealed deficits in the recognition and representation (attribution of a specific affective mental state) of emotion but no particular impairment in the regulation (management of one's emotion) [13]. A later study also confirmed that PD patients have a decreased ability to detect emotion in others, which is associated with difficulty understanding another's point of view (decreased empathy) and has important implications for caregiver burden [14]. Further research confirmed that some of these behavioral impairments are independent of cognitive function yet dependent on dopaminergic availability [15]. Lastly, emotion regulation impairment in the context of appropriate recognition and experience has also been informed [16].

Despite the feeling aspect being an essential element to the concept of emotion, specific components of affective processing such as subjective emotional experience have received less attention in PD. Although previous research indicates that the attributes of subjective experience associated with emotion are preserved in PD [4,5,10,16–18], others have found that PD patients exhibit reduced ratings of subjective arousal in response to aversive or highly arousing pictures [9,19], as well as increased arousal ratings of emotional words reflecting "calm", along with impaired valence ratings for words associated with positive and negative emotions [20]. Albeit the current evidence that the PD process does affect the subjective dimension of emotional experience, the limited existing studies have yielded inconsistent results. Thus, it is still unclear to what extent the ability to feel emotions is compromised in PD.

The diversity of non-motor features that manifest across the spectrum of patients with PD may associate with deficits in the subjective experience of affect, thus accounting for the vast heterogeneity of results produced in previous investigations. For instance, the autonomic nervous system, regarded as one of the earliest components to be involved in the pathological process of PD [21], plays a significant role in different aspects of emotional processing, including its subjective experience [22]. Still, even though distinct neural substrates of autonomic dysfunction overlap with brain circuitry involved in emotional processing [23,24], to the best of our knowledge, the association between autonomic complaints and subjective feeling of emotion in patients with PD has not been systematically addressed. Therefore, in the present study, we aimed to evaluate the subjective emotional experience of PD patients and its relationship with autonomic symptoms and other relevant non-motor manifestations.

PD is a complex disorder with wide variability in the clinical phenotype that suggests the existence of several subtypes of the disease [25]. Traditionally, two major motor clinical phenotypes have been emphasized, but little is known regarding the non-motor variability of PD. Therefore, exploring the link between the affective and autonomic dimensions may contribute

to identifying sets of non-motor features that may be useful in delineating possible non-motor subtypes. Focusing on groups of patients who share underlying pathological mechanisms is crucial for disease-modifying therapies to succeed and achieve precision medicine in PD. Also, the affective symptomatology of PD is costly from economic, social, and personal standpoints [26]. Thus, recognizing and managing emotional dysfunction is essential in providing optimal care to patients with PD.

## Materials and methods

### Participants

We actively recruited 31 idiopathic PD patients and 17 healthy controls at the International Center for Neurological Restoration (CIREN) and the Cuban Center for Neurosciences (CNEURO) (Havana, Cuba), respectively. A movement disorder specialist confirmed the diagnosis of PD. Only participants with no history of neurological disorders other than PD were included. Two patients met the criteria for dementia after clinical evaluation, therefore being excluded from further analysis. One patient reported incomplete data and thus was eligible only for specific analysis. Patients were tested on their usual dosage of dopamine therapy and any other regular treatment. The study protocol is in accordance with the Declaration of Helsinki and was approved by the Research Ethics Committee of CNEURO. Written informed consent was obtained from all PD patients. As the research involved minimal risk to participants, healthy volunteers provided oral informed consent, which was documented by including in the study records the consent script, participant's name, and date consent was obtained.

### Clinical evaluation

Clinical evaluation of patients included the Beck Depression Inventory-II (BDI-II) (cut-off value for depression > 13) [27], the Montreal Cognitive Assessment (MoCA) scale (cut-off values: mild cognitive impairment < 26, dementia < 18) [28], the Rapid Eye Movement (REM) Sleep Behavior Disorder Screening Questionnaire (RBDSQ) (cut-off value for probable RBD > 5) [29], and the Scales for Outcomes in Parkinson's Disease-Autonomic (SCOPA-AUT). Higher scores indicate more severe symptoms for all clinical scales except for the MoCA test. Certain MoCA items were combined, considering some of the previous recommendations [30], to obtain specific cognitive domain index scores: Executive index (trail-making, letter F fluency, and abstraction), Visuoconstructive index (cube copy and clock drawing), Visuospatial index (cube copy, clock drawing, and naming), Attention/Working memory index (digit span, letter A taping, serial 7 subtractions, and sentence repetition) and Memory index (delayed free recall and spatiotemporal orientation). To account for the two cognitive syndromes described in PD [31], we combined the index scores of Executive function and Attention/Working memory, to reflect deficits that are mainly associated with a "fronto-striatal" syndrome, and the index scores of Memory and Visuospatial skills, to identify deficits typically indicative of a "posterior cortical" syndrome. We calculated the ratio of thermoregulatory (TR) and pupillomotor (PM) to gastro-intestinal (GI) and cardiovascular (CV) SCOPA-AUT scores, *i.e.*, (TR + PM)/(GI + CV), to account for a possible heterogeneous pattern of autonomic dysfunction as previously reported [32,33]. The ratio GI/Urinary score was also determined to account for variations in the relative contribution of these complaints to overall autonomic impairment.

### Emotion elicitation protocol

A set of 14 Spanish-dubbed film excerpts, previously assessed for their effectiveness in eliciting emotions in an experimental context and later validated for affective research in a Spanish

**Table 1. Film excerpts for emotion elicitation.**

| Target emotion | Clip | Film | Scene description | Length (min:s) |
|---|---|---|---|---|
| **Disgust** | 1 | The Dentists | *A woman with her tongue cut tries to warn a man that he is in danger.* | 00:56 |
| | 2 | Trainspotting 2 | *A boy rummages inside a filthy toilet.* | 01:41 |
| **Amusement** | 3 | When Harry Met Sally | *Sally fakes an orgasm in a cafeteria.* | 02:45 |
| | 4 | There's Something About Mary 1 | *Ben Stiller wrestles with a small dog.* | 03:02 |
| **Anger** | 5 | American History X | *A neo-Nazi kills a man by crushing his head.* | 01:18 |
| | 6 | In the Name of the Father | *The protagonist is forced to confess under threats and violence.* | 03:34 |
| **Fear** | 7 | The Blair Witch Project | *Following Josh's screams, Heather and Mike stumble upon an abandoned building in the woods. Mike presumably dies.* | 03:57 |
| | 8 | Scream 2 | *Persecution in an institute.* | 03:34 |
| **Neutral** | 9 | Blue 2 | *A man orders papers, and a woman walks around a garden.* | 00:40 |
| | 10 | Blue 3 | *A woman goes up on an escalator.* | 00:24 |
| **Tenderness** | 11 | Dead Poets Society 2 | *A group of students climb to the desks to manifest their solidarity with a fired teacher.* | 02:40 |
| | 12 | Ghost | *A woman sensuously molds wet clay on a pottery wheel, while her shirtless boyfriend sits behind and begins kissing her. They end up making love (the romantic song "Unchained Melody" is played)* | 03:29 |
| **Sadness** | 13 | Schindler's List 1 | *Jews are digging up bodies and carrying them to a moving ramp that dumps them in with a pile of burning corpses. A cart passes by with a tray carrying children corpses.* | 01:18 |
| | 14 | Dead Man Walking | *The main character is put to death by lethal injection. He apologizes for his crime and tells the nun struggling to help him that he loves her.* | 06:40 |

population [34,35], was used to induce Amusement, Anger, Disgust, Fear, Sadness, Tenderness, and Neutral State (Table 1). We selected the film clips expected to elicit the strongest emotional response in native Spanish speakers, as quantified by a weighted average score considering the subjective emotional arousal (weight: 0.75) and intensity (weight: 0.25) values reported for each movie fragment [35]. Videos were played on a laptop and projected on a 42-inch screen while participants were seated comfortably on a sofa placed 2 m from the TV. For each target emotion, two clips were displayed to ensure that the emotional experience was independent of movie content. Two short films were screened to train participants after they had been instructed, and then the experimental clips were randomly presented to make sure that two consecutive excerpts did not target the same emotion. For at least two minutes of recovery time to baseline following each clip, a blank screen was shown. During this period, participants filled in a self-report emotion questionnaire, where they had to: i) tag the experienced emotion by selecting it from the list of target emotions, or either indicate no emotional experience (Neutral State); ii) rate the intensity of the experienced emotion on a 7-point Likert scale; iii) specify (type) any other emotion being experienced with equal or greater intensity; and iv) rate the emotional valence on a 9-point Likert scale self-assessment manikin (SAM). The emotion intensity for the Neutral State was assumed to be the lowest value on the scale (no emotion intensity). Participants were tested individually. The total time length of the protocol was nearly 1.5 h. Physiological and behavioural data not analyzed in this study were also collected.

## Statistical analysis

Statistical analysis was performed using Statistica 10 (StatSoft, Inc., Tulsa, OK, USA). The statistical significance for all tests was set at $p$-value $< 0.05$.

**Descriptive statistics.** Emotion occurrence was assessed in terms of concordance between the target and the experienced emotions (concordance = 1, if the target emotion was

experienced; or 0 otherwise). For each participant, self-reported emotion intensity and valence scores were averaged over the two clips from the same target emotion. Distribution normality was assessed using a Kolmogorov-Smirnov test. Descriptive statistics for continuous data were expressed as mean ± standard deviation (SD) or median (interquartile range), whereas categorical data were expressed as numbers and percentages. Outliers, defined as scores beyond 1.5 interquartile ranges from the first and third quartile of data distribution, were excluded.

**Experimental testing.** Concordance and sex distribution were contrasted between groups using 2 x 2 contingency tables, considering Yates' correction. Between-group comparisons of the remaining variables were assessed through a primary one-way analysis of variance (ANOVA) or a Mann-Whitney U test, depending on data characteristics. Likewise, complementary within-group contrasts were evaluated by a repeated-measures ANOVA or a Friedman ANOVA, following a Fisher's Least Significant Difference (LSD) test or a Wilcoxon matched-pairs test, respectively, for *post hoc* analysis. The potential confounding effects of participant's sex, education level, psychotropic medication, and depression status over the self-reported emotion scores were further explored through regression analysis.

**Correlation analysis.** The primary correlation analyses assessed the association between all self-reported scores of the target emotions that suggested an emotional impairment in PD and the nine scores describing the mood (depression), sleep, cognitive, urinary, gastrointestinal, cardiovascular, pupillomotor, thermoregulatory and sexual domains; comprising overall 54 tests. Pearson *r* or Spearman *R* correlation coefficients were determined and considered meaningful if $r$ or $R \geq 0.45$. The statistical significance was adjusted for multiple comparisons by controlling for the false discovery rate at $q$-value $< 0.1$, using the Benjamini-Hochberg (BH) correction. *Post hoc* analysis exploring the association between the variables from significant tests with the SCOPA-AUT scale, MoCA items, or other emotion scores was also considered.

**Cluster analysis.** Identifying the relevant correlations enabled appointing a set of features for clustering analysis (k-means) to further explore the non-motor correlates of emotional deficits in PD patients. Thus, features that suggested an impaired emotional experience and their most relevant correlates were used as input for the cluster analysis. This technique allows the exploration of the naturally occurring groups (clusters) within a dataset without any predefined labels or classes, as insights are derived from the data. To identify the optimal number of clusters, we applied the elbow method, based on the average degree of distortion measured in terms of Euclidean distances, combined with a recent approach proposed to unambiguously distinguish the elbow point by determining the cosine of interaction angle [36]. A multivariate one-way analysis of covariance (MANCOVA) and Fisher's LSD tests, exploring other variables not considered in the clustering solution with disease duration and disease severity as covariates, were conducted to examine the veracity of the generated subgroups. The effects of participant's sex, age, and education level on the contrasts of interest were also verified. To identify the specific nature of the autonomic and cognitive disturbances, we carried out univariate comparisons of the clinical subscales between the relevant clusters previously identified with MANCOVA *post hoc* analysis. The p-values of the multivariate model and the univariate tests for contrasting motor laterality (%), depression (%), RBD (%) and mild cognitive impairment (%) were adjusted for multiple comparisons using the BH correction (five tests, $q$-value $< 0.1$).

## Results

### Participant demographic and clinical characteristics

Since age and sex have been previously informed to be significant predictors of subjective feeling [37,38], the effect of PD on subjective emotional experience was assessed in a subset of the

**Table 2. Demographic and clinical characteristics of participants.**

| Feature | PD | Control | F/ ($\chi^2$) | $p$ |
|---|---|---|---|---|
| n | 18 | 17 | - | - |
| Sex (male:female) | 13:5 | 6:11 | (3.43) | 0.06 |
| Age (yrs) | 57.1 ± 7.2 | 55.0 ± 4.2 | 1.11 | 0.30 |
| Education level | | | | |
| Low (≤ 9 yrs) | 11 [2] | 0 | - | - |
| Medium (10–13 yrs) | 61 [11] | 24 [4] | - | - |
| High (≥ 14 yrs) | 28 [5] | 76 [13] | **(6.46)** | **0.01** |
| BDI-II (mean ± SD) | 7.4 ± 8.1 | 5.3 ± 6.8 | 0.60 | 0.45 |
| BDI-II (%) [n] | 22 [4] | 15 [2] | (0.00) | 0.99 |
| BDI-II, available n | 18 | 13 | - | - |
| Psychotropic medication | 5 | 1 | (1.61) | 0.20 |
| Anxiolytic | 2 | 1 | - | - |
| Antipsychotic | 1 | 0 | - | - |
| Antidepressant | 1 | 0 | - | - |
| Combination | 1 | 0 | - | - |

Values are expressed as mean ± standard deviation or percentage (number of subjects). ANOVA F or Pearson/Yates corrected $\chi^2$ with associated $p$-value are presented and appear in bold style for significant differences ($p < 0.05$). PD: Parkinson's disease; n: Number of cases; BDI-II: Beck Depression Inventory-II score.

PD patients (n = 18) and a group of healthy controls (n = 17) more homogeneous in terms of age and sex distribution (Table 2). More control than PD participants had completed university, however. The association between subjective emotional feeling and non-motor features was assessed including all individuals with PD (n = 28). The clinical characteristics of patients are summarized in (Table 3). The PD group included members with mild to moderate disease, although most participants (62%, n = 18) had early PD (Hoehn & Yahr stage II). Nine patients (32%) were taking psychotropic drugs, namely anxiolytics (clonazepam: n = 6, alprazolam: n = 1), antipsychotics (quetiapine: n = 1, olanzapine: n = 1), and antidepressants (amitriptyline: n = 2). Except for one patient taking amitriptyline, the intake of psychotropic drugs was separated from the moment of the test for approximately 12 hours. Only one control participant was taking psychotropic medication, specifically alprazolam. All patients complained of autonomic symptoms, and most of them manifested evidence of mild cognitive impairment. The most common autonomic complaints emerged in the urinary and gastrointestinal (GI) domains, with the urinary subscale having the highest contribution to the overall autonomic score. Clinical scores were not significantly associated with age, age at onset, disease duration, or levodopa equivalent daily dose (LEDD) ($p > 0.10$).

### Effect of Parkinson's disease on subjective emotional experience

**Occurrence.** The battery of film excerpts elicited all target emotions in the study participants, except Anger (concordance < 50% in both patient and control groups; Neutral State and Sadness were mostly reported instead). Several participants reported experiencing Tenderness along with Sadness. PD patients experienced Tenderness (53%) significantly less often than controls (Fig 1) (group effect; Pearson $\chi^2$ = 8.15, $p$ = 0.004) and frequently showed no emotional reactivity (Neutral State, 19%) to stimuli recreating tender scenarios. Within the PD group, difficulty in feeling Tenderness occurred regardless of participant's sex, with 80% of females and 69% of males exhibiting some degree of impairment in the experience of Tenderness (mean concordance < 1; Yates corrected $\chi^2$ = 0.02, $p$ = 0.90). Among the patients, 20% of

**Table 3. Clinical characteristics of Parkinson's disease patients.**

| Clinical feature | PD group (correlational analysis) | PD subgroup (experimental analysis) |
|---|---|---|
| n | 28 | 18 |
| Age | 62.4 ± 9.0 | 57.1 ± 7.2 |
| Disease duration (yrs) | 7.6 ± 4.9 | 6.7 ± 4.2 |
| Hoehn & Yahr stage | 2.0 (0.0)[a] | 2.0 (0.5)[b] |
| LEDD (mg) | 745.4 ± 337.8[c] | 694.4 ± 357.4 |
| Psychotropic drugs (%) [n] | 32 [9] | 28 [5] |
| Anxiolytics (%) [n] | 25 [7] | 17 [3] |
| Antipsychotics (%) [n] | 7 [2] | 6 [1] |
| Antidepressants (%) [n] | 7 [2] | 11 [2] |
| Depression (%) [n] | 18 [5] | 22 [4] |
| RBD (%) [n] | 46 [13] | 56 [10] |
| MCI (%) [n] | 68 [19] | 67 [12] |
| Autonomic symptoms (%) [n] | 100 [28] | 100 [18] |
| BDI-II | 7.4 ± 7.0 | 7.4 ± 8.1 |
| RBDSQ | 5.7 ± 2.4 | 6.4 ± 2.6 |
| MoCA | 23.6 ± 2.8 | 23.8 ± 3.0 |
| SCOPA-AUT | 18.2 ± 10.5 | 15.7 ± 9.8 |
| Gastrointestinal | 5.9 ± 3.7 | 4.9 ± 3.2 |
| Urinary | 6.6 ± 4.6 | 5.6 ± 4.7 |
| Cardiovascular | 1.6 ± 2.0 | 1.4 ± 1.9 |
| Thermoregulatory | 2.0 (4.0) | 0.0 (4.0) |
| Pupillomotor | 0.0 (1.0) | 0.0 (1.0) |
| Male sexual | 1.6 ± 1.8[d] | 1.4 ± 1.6[e] |

Values are expressed as mean ± standard deviation or median (interquartile range). Higher scores indicate more severe symptoms for all clinical scales except the MoCA test.

[a] n = 25

[b] n = 16

[c] n = 27

[d] n = 19

[e] n = 12. PD: Parkinson's disease; n: Number of cases; LEDD: Levodopa equivalent daily dose; RBD: Probable Rapid eye movement sleep behavior disorder; MCI: Mild cognitive impairment; BDI-II: Beck Depression Inventory-II; RBDSQ: Rapid eye movement sleep Behavior Disorder Screening Questionnaire; MoCA: Montreal Cognitive Assessment scale; SCOPA-AUT: Scales for Outcomes in Parkinson's Disease-Autonomic.

females *versus* 23% of males did not experience Tenderness at all (mean concordance = 0; Yates corrected $\chi^2 = 0.24$, $p = 0.62$). Instead, some of these patients reported having felt Sadness (11%), Amusement (11%) or Anger (6%) in response to tender scenes, which differed significantly from the overall rate of emotional responses other than Tenderness reported by the healthy controls in the same affective setting, excluding the Neutral State (34% in the PD group *vs.* 13% in the control group; Pearson $\chi^2 = 4.16$, $p = 0.04$). For instance, no healthy participant experienced feelings of Anger in response to the stimuli intended to elicit Tenderness. On the whole, patients reported a higher rate of no emotional experience (Neutral State) when exposed to an affective context (22% in the PD group *vs.* 11% in the control group; Pearson $\chi^2 = 8.30$, $p = 0.004$). The affective protocol was significantly more effective in eliciting feelings of Amusement in the study participants than those of any other emotion (target emotion effect; Friedman ANOVA $\chi^2 = 13.71$, $p = 0.008$). In the healthy participants, sad films mostly elicited pure or mixed feelings of Sadness (62%) and Anger (53%); disgusting films mainly elicited

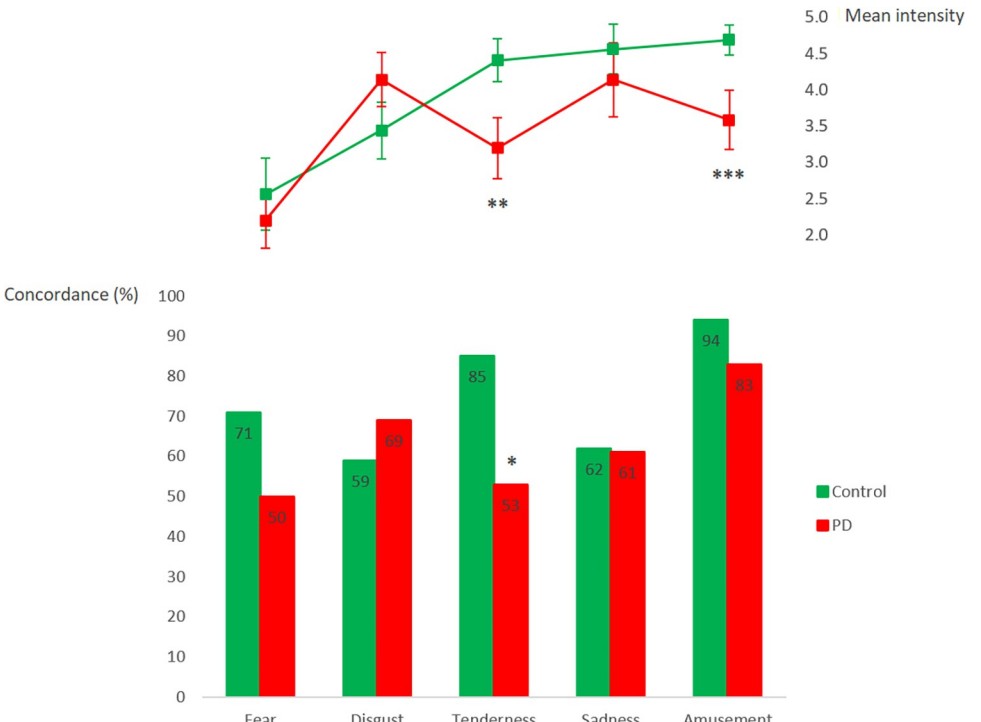

**Fig 1. Emotion self-reported scores in the Parkinson's disease patients (n = 18) and healthy participants (n = 17).**
Emotion concordance (bottom panel) is expressed as the percentage of individuals that did experience the intended
target emotion. Emotion intensity (top panel) ranges between 0 and 6. Error bars represent the standard error of the
mean. *Pearson $\chi^2$ = 8.15, $p$ = 0.004; **One-way ANOVA F = 5.33, $p$ = 0.03; ***One-way ANOVA F = 5.49, $p$ = 0.03.

Disgust (59%) or Fear (26%); whereas scary films mostly elicited either Fear (71%) or Neutral
state (26%).

**Intensity.**   In general, the mean emotion intensity rated by the PD patients remained
below the values reported by the healthy participants across the target emotions (group effect;
multivariate one-way ANOVA F = 2.93, $p$ = 0.03) and was significantly reduced for the experi-
ence of Tenderness and Amusement (Fig 1). In a complementary analysis, we verified that par-
ticipants reported lower intensity for the subjective experience of Fear than for any other
emotional experience (target emotion effect; 2 x 5 repeated measures ANOVA F = 10.05,
$p$ < 0.0001). PD patients reported the strongest feelings for Disgust, which scored among the
least intense emotional experiences in the control participants. Yet, the target emotion and
group interaction effect was not statistically significant (2 x 5 repeated measures ANOVA
F = 2.24, $p$ = 0.07).

**Valence.**   Emotional valence ratings were not significantly different between groups
(group effect; multivariate one-way ANOVA F = 0.92, $p$ = 0.50), although pleasant emotions
(6.1 ± 1.2 in the PD group *vs*. 6.7 ± 0.9 in the control group; ANOVA F = 3.69, $p$ = 0.06) exhib-
ited a greater variation between groups than unpleasant emotions (1.5 ± 1.0 in the PD group
*vs*. 1.4 ± 0.7 in the control group; ANOVA F = 0.12, $p$ = 0.74). In a complementary analysis, we
confirmed that the self-reported emotional valence for the positive and negative target emo-
tions was significantly higher and lower than for the Neutral State (target emotion effect; 2 x 6
repeated measures ANOVA F = 145.99, $p$ < 0.0001). The target emotion and group interaction
effect was not statistically significant (2 x 6 repeated measures ANOVA F = 1.10, $p$ = 0.36).

**Potential confounders.**   Multiple regression analysis was applied to consider the potential
contribution of relevant confounders of subjective emotional experience. After controlling for

the effect of the sex of participants ($\beta 1$) from the experimental and control groups, we only confirmed a significant impact of PD ($\beta 2$) on the occurrence of Tenderness ($t_{\beta 1}$ = -0.09, $p$ = 0.93; $t_{\beta 2}$ = -2.43, $p$ = 0.02) and the intensity of both Tenderness ($t_{\beta 1}$ = 0.11, $p$ = 0.92; $t_{\beta 2}$ = -2.12, $p$ = 0.04) and Amusement ($t_{\beta 1}$ = 1.57, $p$ = 0.13; $t_{\beta 2}$ = -2.77, $p$ = 0.009). In addition, we also confirmed that among these participants, the level of education did not significantly explain the differences in Tenderness occurrence (t = 0.76, p = 0.45), Tenderness intensity (t = -0.31, p = 0.76), and Amusement intensity (t = -0.07, p = 0.94). Furthermore, depression ($\beta 1$) and psychotropic drug medication ($\beta 2$) in the PD patients did not show to have a significant influence on MoCA score ($t_{\beta 1}$ = 0.75, $p$ = 0.46; $t_{\beta 2}$ = 1.13, $p$ = 0.27), Tenderness occurrence ($t_{\beta 1}$ = 1.15, $p$ = 0.26; $t_{\beta 2}$ = -0.45, $p$ = 0.66) or intensity of Tenderness ($t_{\beta 1}$ = 0.24, $p$ = 0.81; $t_{\beta 2}$ = -0.98, $p$ = 0.34) and Amusement ($t_{\beta 1}$ = -0.44, $p$ = 0.67; $t_{\beta 2}$ = -0.77, $p$ = 0.45). Finally, after controlling for the effects of participant's sex and age where significant, the impact of PD on Amusement intensity (t = -2.30, $p$ = 0.03) and Tenderness occurrence (t = -2.09, $p$ = 0.04) was also confirmed in the whole sample of the study participants (n = 47), whereas the differences in Tenderness intensity were significantly explained only by the occurrence of this emotion (t = 2.14, $p$ = 0.04).

### Association between subjective emotional ratings and clinical features

**Correlation analysis.** We assessed the correlations between the clinical scales and the scores of occurrence, intensity, and valence reported for the two target emotions that suggested an impaired subjective feeling, *i.e.*, Amusement and Tenderness, in the whole group of PD patients (n = 28, Table 4). The intensity and valence of the Amusement subjective experience were significantly correlated with the prevalence of urinary symptoms. *Post hoc* analysis also confirmed a significant correlation between the valence reported during the Amusement experience and the total SCOPA-AUT score. The valence reported during the experience of Tenderness was significantly correlated with the prevalence of sexual dysfunction in male patients with PD (n = 19). *Post hoc* analysis also revealed that the men's sexual symptom prevalence was independently associated with the valence reported during the experience of Disgust. The occurrence of Tenderness was significantly associated with the MoCA scale and the prevalence of GI symptoms. *Post hoc* analysis also confirmed significant correlations with the total SCOPA-AUT score, and the Attention and Visuospatial/Executive items of the MoCA scale. BDI-II and RBDSQ showed no association with subjective emotional ratings. Excluding the four patients under antidepressant and antipsychotic treatment from the analysis did not alter these results, confirming no significant effect of these psychotropic drugs on the present findings.

**Table 4. Correlations between subjective emotional ratings and clinical scores in the Parkinson's disease group.**

| Target emotion | Self-report score | SCOPA-AUT | | | | MoCA | | |
|---|---|---|---|---|---|---|---|---|
| | | Total | Urinary | GI | Sexual (men) | Total | Attention | Visuoexecutive |
| **Amusement** | Intensity | -0.26 | **-0.48** | 0.004 | -0.18 | -0.06 | 0.20 | -0.01 |
| | Valence | **-0.51** | **-0.49** | -0.42 | -0.28 | -0.07 | 0.18 | -0.03 |
| **Tenderness** | Occurrence | **-0.45** | -0.32 | **-0.55** | -0.11 | **0.61** | **0.50** | **0.47** |
| | Valence | -0.40 | -0.43 | -0.26 | **-0.70** | 0.18 | 0.20 | 0.31 |
| **Disgust** | Valence | 0.03 | 0.16 | -0.04 | **0.61** | -0.14 | -0.003 | -0.18 |

Strong correlations (Pearson or Spearman coefficient $\geq$ 0.45) in the Parkinson's disease group (n = 28 patients, n = 19 males) that remained significant (*p*-value < 0.05) after adjusting for multiple comparisons by controlling for the false discovery rate at *q*-value < 0.1 appear in bold red style. Besides the MoCA items specified, Tenderness occurrence also correlated significantly with the Attention and Visuospatial indexes (Spearman R = 0.39 and R = 0.41, respectively). Higher and lower scores indicate more severe symptoms for the SCOPA-AUT and MoCA scales, respectively. GI: Gastrointestinal; MoCA: Montreal Cognitive Assessment scale; SCOPA-AUT: Scales for Outcomes in Parkinson's Disease-Autonomic.

The intensity and valence scores reported for the experience of Amusement and Tenderness were not significantly correlated (r = 0.32, *p* = 0.12 and r = 0.02, *p* = 0.93; respectively). Overall, emotion scores were not significantly correlated with the age at disease onset, disease duration, disease stage, LEDD, or levodopa equivalent dose at the time of the test (LED). Only the occurrence, intensity, and valence of Disgust were significantly correlated with the LEDD or LED (*e. g.*, occurrence: LEDD R = -0.47, LED R = -0.41; intensity: disease duration r = -0.39, LEED r = -0.40; valence: LEDD R = 0.43, LED R = 0.47).

**Cluster analysis.** To further explore the clinical correlates of disturbances in subjective emotional experience in PD patients, we carried out a k-means cluster analysis, including as input the specific features suggesting emotional impairment, *i.e.*, Amusement intensity and Tenderness occurrence, and the non-motor aspects most strongly associated with them according to the correlation analyses, *i.e.*, those from the autonomic (urinary SCOPA-AUT score) and cognitive (MoCA score) domains, resulting in four input variables. As shown in Table 5, the clustering approach allowed us to identify five patient profiles with significant differences in the affective, autonomic, cognitive, mood, and sleep dimensions (MANCOVA Wilks F = 6.24, *p* = 0.009). Patients in cluster 1, exhibiting an impaired feeling of both Amusement and Tenderness, displayed a generalized non-motor dysfunction with a more significant contribution of autonomic and cognitive domains. In contrast, impairments in the other clusters were more domain-specific with substantial involvement of mood and autonomic function in cluster 2 (Amusement selectively impaired), subjective feeling in cluster 3 (impaired feeling of Amusement, Tenderness, and other emotions), and sleep and cognition in cluster 4 (Tenderness selectively impaired). Patients in cluster 5 exhibited preservation of all the functions assessed and were receiving the lowest dose of antiparkinsonian medication (LEDD). The most severe autonomic dysregulation, characterized by prevalent urinary symptoms, emerged in cluster 1, also displaying the most significant GI and cardiovascular impairment, and cluster 2, exhibiting the largest thermoregulatory and pupillomotor dysfunction. These clusters differed significantly in the prevalence of urinary complaints and the relative contribution of the other autonomic domains for which they reported the greatest involvement, *i.e.*, (TR + PM)/(GI+CV). Female patients were distributed among these subgroups with the highest prevalence of urinary symptoms, although differences in sex distribution among clusters were not statistically significant. Patients in cluster 2, who reported the lowest valence of Amusement experience, also exhibited the highest depression rate, a young age at disease onset ($\leq$ 50 years for all members), the most extended disease duration, and received the highest LEDD. Depression and thermoregulation scores tended to be higher in cluster 2, reaching statistical significance when assessed in planned comparisons. Preserved cognition characterized the patients in clusters 2 and 5. Our results further revealed predominant deficits among cognitively impaired participants, namely impaired attention, executive function, and language in cluster 1, visuoconstructive abilities in cluster 3, and memory and visuospatial skills in cluster 4. A prominent "fronto-striatal" cognitive syndrome distinguished the patients in cluster 1 from those in cluster 3, whereas features of the "posterior cortical" syndrome distinguished the patients in cluster 4 from those in cluster 1. Excluding the patients with significant "fronto-striatal" deficits (cluster 1), 83% of the variance (adjusted $R^2$) observed in global cognition (MoCA score) among the PD patients (n = 19) was independently explained by participant's sex (male = 0, female = 1; $\beta$ = -0.25, t = -2.20, *p* = 0.04), the ability to experience Tenderness ($\beta$ = 0.48, t = 4.73, *p* = 0.0003), semantic knowledge ("naming" MoCA item; $\beta$ = 0.26, t = 2.58, *p* = 0.02), and memory ("delayed recall" MoCA item; $\beta$ = 0.48, t = 4.28, *p* = 0.0008), as explored through multiple regression analysis (F = 22.22, *p* < 0.00001, standard error of estimate = 1.06). Disease duration, disease stage, education level, and age were not significantly associated with the global cognitive status of the patients.

**Table 5. Clusters of Parkinson's disease patients profiled according to relevant non-motor disturbances.**

| Feature | Cluster 1 (n = 9) | Cluster 2 (n = 4) | Cluster 3 (n = 6) | Cluster 4 (n = 5) | Cluster 5 (n = 4) | F/(H)/[χ²] | p |
|---|---|---|---|---|---|---|---|
| Non-motor domain predominantly involved | Urinary & Cognition (Attention, Language, and Executive function) | Mood & Thermoregulation | Subjective feeling & Cognition (Visuoconstructive skills) | Sleep & Cognition (Memory and Visuospatial skills) | Absent | - | - |
| Sex (% male) [n] | 56 [5] | 50 [2] | 100 [6] | 100 [5] | 100 [4] | [1.72] | > 0.19 |
| Age (yrs) | 65.9 ± 6.7 | 60.8 ± 8.6 | 64.8 ± 9.4 | 59.4 ± 7.8 | 56.0 ± 13.7 | 1.15 | 0.36 |
| Education, >12 yrs (%) [n] | 77.8 [7] | 100 [4] | 66.7 [4] | 80.0 [4] | 75.0 [3] | [1.67] | > 0.20 |
| Disease duration (yrs) | 7.1 ± 3.8 | 11.5 ± 8.3 | 8.3 ± 4.9 | 6.6 ± 4.3 | 4.8 ± 2.5 | 1.11 | 0.37 |
| Age at onset (yrs) | 58.3 ± 8.2 | 48.8 ± 1.3 | 56.5 ± 12.9 | 52.8 ± 5.8 | 51.3 ± 13.4 | 0.97 | 0.45 |
| Hoehn & Yahr stage | 2.1 ± 0.3 | 2.7 ± 0.6 | 2.3 ± 0.5 | 1.8 ± 0.5 | 2.0 ± 1.0 | 1.53 | 0.23 |
| LEDD | 726.6 ± 359.5 | 878.1 ± 484.6 | 827.1 ± 192.7 | 807.5 ± 304.1 | **512.5 ± 513.9** | 4.08 | **0.01[a]** |
| Motor laterality (right %) [n] | 75 [3] | 67 [2] | 80 [4] | **0** | 67 [2] | [4.80] | **< 0.03[b]** |
| Laterality, available n [%] | 4 [44] | 3 [75] | 5 [83] | 4 [80] | 3 [75] | - | - |
| Psychotropic med. (%) [n] | 33 [3] | 50 [2] | 33 [2] | 20 [1] | 25 [1] | [0.90] | > 0.34 |
| Depression (%) [n] | 11 [1] | **75 [3]** | 0 | 20 [1] | 0 | [4.80] | **< 0.03[c]** |
| RBD (%) [n] | 44 [4] | 0 | 50 [3] | **100 [5]** | 25 [1] | [4.32] | **< 0.04[d]** |
| MCI (%) [n] | **100 [9]** | 0 | 83 [5] | **100 [5]** | 0 | [6.67] | **< 0.01[e]** |
| BDI-II | 9.2 ± 8.8 | 12.8 ± 8.4 | 5.2 ± 4.6 | 6.0 ± 5.5 | 3.0 ± 2.4 | 1.16 | 0.37 |
| RBDSQ | 5.6 ± 2.2 | 3.8 ± 0.5 | 6.3 ± 3.6 | **7.4 ± 1.1** | 4.8 ± 1.3 | **2.91** | **0.04[f]** |
| MoCA | **22.0 ± 2.7** | 27.5 ± 1.3 | **23.5 ± 1.2** | **21.6 ± 1.1** | 26.3 ± 0.5 | 4.05 | **0.01[g]** |
| Naming item | 3.0 (0.0) | 3.0 (0.0) | 3.0 (0.0) | 2.6 ± 0.5 | 2.8 ± 0.5 | (4.64) | 0.33 |
| Language item | **2.0 (0.0)** | 2.8 ± 0.5 | 2.7 ± 0.5 | 2.4 ± 0.9 | 2.5 ± 0.6 | (9.39) | 0.05 |
| Delayed recall item | 2.2 ± 1.3 | 3.3 ± 1.3 | 1.8 ± 1.5 | 1.0 ± 1.0 | 2.8 ± 1.3 | 2.05 | 0.12 |
| Executive index | **2.7 ± 1.3** | 4.0 (0.0) | 3.3 ± 0.8 | 3.4 ± 0.5 | 4.0 ± 0.0 | (8.69) | 0.07 |
| Visuoconstructional index | 2.7 ± 0.9 | 3.8 ± 0.5 | **2.5 ± 0.8** | 2.6 ± 1.1 | 3.0 ± 1.2 | 1.38 | 0.27 |
| Visuospatial index | 5.6 ± 0.9 | 6.8 ± 0.5 | 5.5 ± 0.8 | **5.2 ± 1.6** | 5.8 ± 1.5 | 1.25 | 0.32 |
| Attention index | **5.0 (2.0)** | 7.5 ± 0.6 | 6.5 ± 1.4 | 6.2 ± 0.8 | 7.5 ± 0.6 | (11.58) | **0.02[h]** |
| Memory index | 8.0 ± 1.2 | 9.3 ± 1.3 | 7.7 ± 1.6 | **6.6 ± 0.9** | 8.8 ± 1.3 | 2.86 | **< 0.05[i]** |
| FSS index | **8.1 ± 2.4** | 11.5 ± 0.6 | 9.8 ± 1.6 | 9.6 ± 0.5 | 11.5 ± 0.6 | 4.46 | **0.008[j]** |
| PCS index | **13.6 ± 1.5** | 16.0 ± 1.4 | **13.2 ± 1.5** | **11.8 ± 1.5** | 14.5 ± 1.0 | 5.35 | **0.003[k]** |
| SCOPA-AUT | **27.7 ± 9.6** | **19.5 ± 8.3** | **15.5 ± 3.6** | 12.0 ± 5.2 | 7.5 ± 10.3 | 4.89 | **0.004[l]** |
| Gastrointestinal (GI) | 8.2 ± 4.4 | 5.3 ± 4.3 | 5.2 ± 2.3 | 5.2 ± 1.3 | 3.0 ± 3.6 | 1.83 | 0.16 |
| Urinary | **12.1 ± 2.4** | **6.8 ± 1.5** | **5.2 ± 1.7** | 2.6 ± 1.1 | 1.3 ± 1.9 | 33.5 | **< 0.001[m]** |
| Cardiovascular (CV) | 3.0 ± 2.4 | 0.5 ± 0.6 | 1.7 ± 2.3 | 0.8 ± 1.1 | 0.3 ± 0.5 | 2.37 | 0.08 |
| Thermoregulatory (TR) | 2.6 ± 2.6 | 5.0 ± 4.2 | 1.3 ± 1.5 | 2.2 ± 2.0 | 1.8 ± 3.5 | 1.22 | 0.33 |
| Pupillomotor (PM) | 0.8 ± 1.1 | 1.3 ± 1.5 | 0.3 ± 0.8 | 0.6 ± 1.3 | 0.5 ± 0.6 | 0.47 | 0.76 |
| Male sexual | 2.5 ± 1.7 | 3.0[1] | 1.8 ± 2.2 | 0.8 ± 1.5 | 0.8 ± 1.5 | 0.84 | 0.52 |
| Female sexual | 2.0 (0.0)[2] | 3.0[1] | - | - | - | - | - |
| GI/Urinary | 0.7 ± 0.3 | 0.7 ± 0.5 | 1.1 ± 0.6 | **2.6 ± 1.9** | 3.0 ± 3.2 | 3.17 | **0.03[n]** |
| (TR+PM)/(GI+CV) | 0.3 ± 0.2 | **1.7 ± 1.6** | 0.2 ± 0.2 | 0.4 ± 0.4 | 0.3 ± 0.6 | 3.79 | **0.02[o]** |
| Amusement occurrence | 0.7 ± 0.4 | 0.8 ± 0.5 | 0.7 ± 0.4 | 1.0 ± 0.0 | 1.0 ± 0.0 | 1.06 | 0.42 |
| Tenderness occurrence | 0.5 (0.0) | 0.9 ± 0.3 | **0.5 ± 0.5** | **0.3 ± 0.3** | 0.9 ± 0.3 | 2.99 | **0.04[p]** |
| Sadness occurrence | 0.7 ± 0.4 | 0.8 ± 0.3 | 0.4 ± 0.4 | 0.4 ± 0.4 | 0.9 ± 0.3 | 1.10 | 0.40 |
| Disgust occurrence | 0.6 ± 0.3 | 0.6 ± 0.3 | 0.8 ± 0.4 | 0.7 ± 0.4 | 0.8 ± 0.3 | 0.49 | 0.81 |
| Fear occurrence | 0.6 ± 0.4 | 0.6 ± 0.5 | 0.8 ± 0.4 | 0.4 ± 0.4 | 0.3 ± 0.5 | 1.91 | 0.14 |
| (Tend + Sad) occurrence | **0.5 ± 0.3** | 0.8 ± 0.1 | **0.5 ± 0.3** | **0.4 ± 0.3** | 0.9 ± 0.1 | 2.81 | **0.04[q]** |

(*Continued*)

**Table 5.** (Continued)

| Feature | Cluster 1 (n = 9) | Cluster 2 (n = 4) | Cluster 3 (n = 6) | Cluster 4 (n = 5) | Cluster 5 (n = 4) | F/(H)/ [$\chi^2$] | p |
|---|---|---|---|---|---|---|---|
| Amusement intensity | **2.8 ± 1.5** | **3.3 ± 1.3** | **2.3 ± 1.3** | 5.6 ± 0.9 | 4.9 ± 0.9 | **3.72** | **0.02[r]** |
| Tenderness intensity | 3.9 ± 1.9 | 4.5 ± 1.5 | 3.8 ± 1.4 | 2.9 ± 2.1 | 3.9 ± 1.5 | 0.91 | 0.51 |
| Sadness intensity | 4.2 ± 2.3 | 4.5 ± 0.9 | 4.4 ± 0.9 | 4.7 ± 2.6 | 4.3 ± 1.5 | 0.11 | 0.99 |
| Disgust intensity | 4.1 ± 1.9 | 3.8 ± 1.7 | 4.1 ± 0.5 | 4.9 ± 1.6 | 4.5 ± 1.0 | 0.49 | 0.81 |
| Fear intensity | 2.3 ± 1.8 | 2.0 ± 2.3 | **3.4 ± 1.4** | 3.0 ± 1.1 | 1.5 ± 1.7 | 1.91 | 0.14 |
| Valence[Amusement] | 5.2 ± 1.0 | 5.0 ± 1.7 | 5.3 ± 1.6 | 6.7 ± 1.1 | 6.5 ± 1.0 | 0.91 | 0.51 |
| Valence[Tenderness] | 5.7 ± 1.4 | 6.5 ± 1.1 | 6.6 ± 1.1 | 7.1 ± 1.5 | 6.5 ± 0.9 | 0.77 | 0.61 |
| Valence[Sadness] | 1.3 ± 1.7 | 0.6 ± 0.5 | 0.9 ± 0.8 | 0.6 ± 1.3 | 1.5 ± 1.2 | 0.54 | 0.77 |
| Valence[Disgust] | 1.7 ± 1.3 | 1.0 ± 0.9 | **2.4 ± 2.3** | 0.6 ± 0.9 | 0.9 ± 0.9 | 0.95 | 0.49 |
| Valence[Fear] | 1.9 ± 1.4 | 1.3 ± 1.0 | **2.3 ± 1.7** | 1.4 ± 2.1 | 3.6 ± 1.1 | 1.82 | 0.15 |

Values are expressed as mean ± standard deviation, median (interquartile range), or percentage [number of subjects]. MANCOVA or ANOVA F, Kruskal-Wallis ANOVA H, or Pearson/Yates corrected $\chi^2$ with associated p-value are presented and appear in bold style for significant differences ($p < 0.05$). [1]n = 1; [2]n = 2; n: Number of cases; LEDD: Levodopa equivalent daily dose; RBD: Rapid eye movement sleep behavior disorder; MCI: Mild cognitive impairment; BDI-II: Beck Depression Inventory-II score; RBDSQ: Rapid eye movement sleep Behavior Disorder Screening Questionnaire score; MoCA: Montreal Cognitive Assessment scale score; FSS: Fronto-striatal syndrome; PCS: Posterior cortical syndrome; SCOPA-AUT: Scales for Outcomes in Parkinson's Disease-Autonomic score; GI: Gastrointestinal SCOPA-AUT score; TR: Thermoregulatory SCOPA-AUT score, PM: Pupillomotor SCOPA-AUT score, CV: Cardiovascular SCOPA-AUT score, Tend: Tenderness; Sad: Sadness.

All contrasts for which a significant *cluster* effect was found are described below. For some of these, we additionally found a significant interaction effect between the *cluster* and *participant's sex*, which appears as "[females]" to indicate that we are referring to the female patients in the respective cluster.

[a]1, 2, 3, 4 *vs.* 5.

[b]1, 3 *vs.* 4.

[c]2 *vs.* 1, 3, 5 (excluding 1 in the case of the BDI-II score after controlling also for participant's sex and age).

[d]4 *vs.* 1, 2, 5.

[e]1, 3, 4 *vs.* 2, 5.

[f]4 *vs.* 2.

[g]1, 4 *vs.* 2, 5; 3 *vs.* 2 (similar after controlling also for participant's sex, age, and education level).

[*]1 *vs.* 2, 5 (Attention index, *vs.* 2 [females]; Executive index, *vs.* 2 [females]; "Fronto-striatal" syndrome); 1 *vs.* 2 [females] (Language item).

[*]4 *vs.* 2, 5 (Memory index, *vs.* 2 [females]; "Posterior cortical" syndrome); 4 *vs.* 2 (Delayed recall, *vs.* 2 [females]; Visuospatial index, before controlling for participant's sex and education level).

[*]3 *vs.* 2 (Visuoconstruccional index, before controlling for participant's sex, age, and education level; "Posterior cortical" syndrome).

[h]1 *vs.* 2, 5 (univariate analysis, similar after controlling for disease duration, disease stage, participant's sex, age, and education level).

[i]4 *vs.* 2, 5 (univariate analysis, similar after controlling for disease duration, participant's sex, age, and education level).

[j]1 *vs.* 2, 3, 5 (univariate analysis); 1 *vs.* 2, 5 (after controlling for age and education level).

[k]4 *vs.* 1, 2, 5; 1, 3 *vs.* 2 [females] (univariate analysis; similar after controlling for participant's sex, age, disease duration and education level).

[l]1 *vs.* 3, 4, 5; 2, 3 *vs.* 5 (similar after controlling also for participant's sex and age).

[*]1 *vs.* 3, 4, 5 (Urinary); 1 [females] *vs.* 4, 5 (CV); 1 *vs.* 5 (GI).

[*]2, 3 *vs.* 5 (Urinary); 2 [females] *vs.* 5 (TR, PM).

[m]1 *vs.* 2, 3, 4, 5; 2 *vs.* 4, 5; 3 *vs.* 5 (univariate analysis).

[n]4 *vs* 1, 2 [females], 3 (univariate analysis; similar after controlling for participant's sex and age).

[o]2 [females] *vs.* 1, 3, 4, 5 (univariate analysis; similar after controlling for participant's sex, age, disease duration and disease stage).

[p]1, 4 *vs.* 2, 5; 3 *vs.* 5; not 1 *vs.* 2 after controlling also for participant's sex and age.

[q]1, 3, 4 *vs.* 5; 4 *vs.* 2 (univariate results; excluding 1 after controlling for participant's sex).

[r]1, 2, 3 *vs.* 4; 1, 3 *vs.* 5 (no effect of depression); not 1 *vs.* 5, after controlling also for participant's sex and age.

It is worth noting that despite the relatively high occurrence of Sadness in the cognitively intact PD patients (*e.g.*, > 80% in clusters 2 and 5), sad feelings occurred less often among participants in clusters 3 and 4, and remained below the study threshold for considering that the

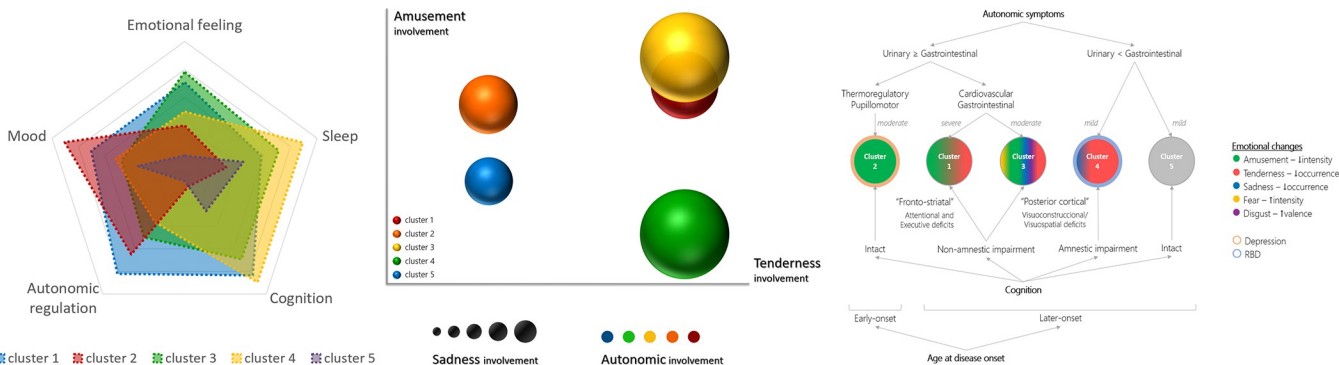

**Fig 2. Dominant non-motor characteristics in the clusters of patients with Parkinson's disease.** (A) Relative involvement of the non-motor domains (higher score represents greater impairment). The domain of emotion summarizes the deficits in Amusement intensity, Tenderness occurrence, and Sadness occurrence. (B) Delineation of PD cluster across the feature space representing the involvement of Amusement intensity *vs.* Tenderness occurrence; the involvement of Sadness occurrence and autonomic manifestations is represented by the relative size and color of bubbles, respectively. The graphs show the standardized mean or median scores (n = 28 patients). The inverse value was presented in the case of those features for which a greater score denotes a better status. c1-c5: Cluster 1—cluster 5. (C) Deficits in subjective emotional experience across autonomic and cognitive profiles. Autonomic symptoms with a substantial relative contribution and cognitive domains with more prominent impairments are illustrated. RBD: Rapid eye movement sleep behavior disorder.

target emotion had been elicited ($< 50\%$). To account for the similarity between Tenderness and Sadness in terms of affiliative social meaning, we further evaluate the occurrence of these "affiliative emotions", finding that cognitively impaired patients experienced affiliative feelings significantly less often than cognitively intact individuals, especially those in clusters 3 and 4 after controlling for participant's sex. Before correcting for multiple comparisons, and after controlling for disease duration, disease severity, and LEDD, patients in cluster 3 reported higher valence for the feelings of Disgust ($p = 0.02$) and Fear ($p = 0.03$) compared to those in cluster 4 and higher intensity of Fear compared to members of cluster 2 ($p = 0.03$) and cluster 5 ($p = 0.02$). This trend remained when patients of cluster 3 where compared to control participants for the valence (control group; Disgust = 1.3 ± 1.5, Fear = 2.2 ± 1.2) and intensity (control group; Fear = 2.6 ± 2.1) self-report scores. Among the male participants with PD, the occurrence of affiliative emotions and the intensity of Fear were significantly correlated (r = -0.46). In general, patients in cluster 3 presented intermediate scores in the non-motor symptom assessment, yet they reported the most significant deficits in subjective emotional experience (Fig 2). Fig 2 summarizes the contribution of each non-motor domain to the overall symptom complex outlining each cluster. Among the patients with prominent clinical manifestations, those in cluster 2 (early-onset PD, moderate dysautonomia, high rate of depression, intact cognition, no RBD, Amusement selectively impaired) and cluster 4 (mild dysautonomia, low rate of depression, amnestic cognitive impairment, prevalent RBD, Tenderness selectively impaired) exhibited the most contrasting non-motor symptom profile.

## Discussion

We assessed the subjective experience of emotion in PD patients compared to healthy controls and explored the association between subjective emotional experience and non-motor manifestations. We used a battery of film excerpts to elicit Amusement, Anger, Disgust, Fear, Sadness, Tenderness, and Neutral State. The subjective emotional experience was evaluated by analyzing the self-report scores of the emotion category (concordance), the intensity of the experienced emotion, and the emotional valence. Validated clinical scales were administered to assess the prevalence of non-motor symptoms of autonomic dysfunction, cognitive impairment, REM sleep disturbance, and depression. All target emotions were elicited except

Anger, which is challenging to induce reliably in a lab-controlled environment, and for which it has been suggested that induction methods involving personal contact lead to a deeper activation [4,39]. We found in the PD group a significant reduction of the occurrence of Tenderness and the mean (over the two excerpts) emotion intensity reported in response to the film clips targeting Tenderness and Amusement. The subjective intensity and the emotional valence reported in response to stimuli targeting Amusement were associated with urinary symptom prevalence; the emotional valence reported for the excerpts targeting Tenderness and Disgust was associated with men's sexual symptom prevalence, whereas Tenderness occurrence was associated with both the cognitive and the autonomic domains, mainly with global cognition and GI symptom prevalence. Overall, emotion scores did not significantly correlate with disease duration, suggesting that emotional dysfunction may be an integral part of the initial motor phase of PD [40]. We identified five clusters of PD patients based on relevant affective, autonomic, mood, sleep, and cognitive deficits, which differed significantly in their non-motor profile.

## Changes in the subjective emotional experience of Tenderness and Amusement

The subjective emotional experience of Tenderness and Amusement was found to be compromised in the PD patients. These deficits were not found to be related to an impaired valuation of hedonic tone (emotional valence). In keeping with previous investigations using a dynamic presentation of stimuli in ecologically valid scenarios, we found no significant impairment in the subjective experience of the basic emotions [4,17,18], which suggests that it is the subjective feeling of complex affect what may be selectively impaired in PD. Basic emotions are mainly characterized by feeling aspects, whereas complex emotions require further elaboration, and therefore the cognitive content is an essential constituent [41]. Emotional complexity, a relevant dimension for encoding feelings, arises from affective experiences that involve mixed emotions, combined contextual cues, as well as the integration of cognitive and affective components [41,42]. In contrast to our findings, two prior studies using film clips for emotion elicitation reported unimpaired subjective experience of Amusement [5] and Tenderness [18] in PD. In the first study, however, the patients reported lower Amusement intensity levels than the healthy controls, but the differences did not reach statistical significance ($p = 0.06$) [5]. Despite the higher rate of depression in that experimental group, the patient cohorts in both these investigations were cognitively high functioning, with no significant deficits in attention, working memory, or visuoperceptual skills (as assessed by a facial recognition test) as verified in the second study [18]. In fact, a limited capacity for sustained attention and working memory was found to correlate significantly with the reduced ability of PD patients in the present study to experience Tenderness. In addition to the higher prevalence of cognitive symptoms in our patients, the emotion elicitation procedure used in our protocol may have imposed greater cognitive demands, as the film clips lasted longer, generated ambivalent feelings during the experience of Tenderness [35], and involved sexual hints (*e.g.*, sexual Amusement), enhanced with romantic cues in some scenes recreating Tenderness and affection. In agreement with our findings, though, a recent study investigating the ability of PD patients to recognize dynamic facial expressions reported a selective impairment in their capacity for processing cognitively complex emotions [43]. Taken together, the existing evidence supports that some affective deficits in PD are most likely to manifest when more complex and sophisticated cognitive skills for emotional processing are in demand.

Despite the audiovisual material used to elicit Tenderness may have produced a mixed state of Tenderness and Sadness among the study participants [35], our findings support the

possibility that patients unable to experience Tenderness may generate alternative affective responses that demand less cognitive effort or recruit preserved affective territories. Along these lines, 17% of patients reported Sadness or Anger in response to the film excerpts intended to elicit Tenderness. These non-target categories represent dimorphous emotions that communicate positive affective states when displayed in positive contexts that represent emotionally evocative situations [44]. Such a de-complexification of the emotional response may arise from the integration of the limited cognitive resources available to process affect as a coping strategy that compensates for the deficits associated with the experience of Tenderness. Consistent with this finding, increased subjective experience of non-target emotions in response to film clips has been reported in patients with neurodegenerative diseases reflecting alterations in emotional processing [45]. On the other hand, a restriction or diminution in the intensity or frequency of emotions may be linked to the presence of mood disorders that are often comorbid in PD or to a side effect of medication. Concerning the affective motivation, another basic dimension of human affect, Tenderness, in contrast to Amusement, can be regarded as a high-approach motivational emotion [46,47]. Thus, a reduced occurrence of Tenderness may associate with a reduction of goal-directed behavior, a symptom of apathy. In contrast, a diminished intensity of both positive emotions may relate to a lowered ability to experience pleasure (anhedonia), a symptom of depression [48]. Still, in the present study, no significant effect of depression or psychotropic/dopaminergic medication was verified to explain the variance in participants' ratings of emotional intensity and valence, similar to what has been previously reported [17,49]. These results suggest that the experience of Tenderness and Amusement may further depend on systems of neurotransmitters other than dopamine, compatible with models which propose that different contributions of neuromodulators, *i.e.*, dopamine, noradrenaline, serotonin, and acetylcholine, result in the more complex "higher-order" emotions [50,51]. In fact, equally high levels of noradrenaline and dopamine are found in the nucleus accumbens [52], suggesting that the noradrenergic system is implicated as well in mediating reward and motivation. In keeping with this, degeneration of both the locus coeruleus, the principal source of noradrenaline in the brain, and the nucleus accumbens has been related to motivational issues in PD that may be insensitive to dopamine medication [53,54].

Subjective intensity ratings of complex emotions have been found to associate with activation in the right temporoparietal junction, which atrophy may underlie changes in the subjective experience of Tenderness and Amusement [42,55]. Additionally, feeling Amusement implicates the mesolimbic dopaminergic reward system, in which a pronounced amygdala activation has been described during the appreciation phase of humor (the subjective experience of Amusement) [56,57]. Consistently, humor appreciation has been reported to be decreased among PD patients, which may follow from increased inhibition of the mesolimbic pathway secondary to dopaminergic dysfunction [49,58]. In contrast, the impaired experience of Tenderness in patients with PD may be associated with atrophy of basal forebrain (BF) regions, especially the septo-hypothalamic area, and medial frontopolar cortex [59–61]. Tenderness is an empathic emotion that arises in response to a cognitive appraisal of vulnerability [62]. In keeping with this, a reduction of empathy, mainly driven by alterations in the cognitive component, has been informed in PD [63]. Both, Amusement and Tenderness, promote adaptive responses to the potential for reward by facilitating complex behavior that fosters communication and social interaction [64]. Further, the subjective experience of positive emotions is associated with several good health and psychological outcomes [65]. In addition, positive affect can serve adaptive functions by contributing to achieving efficient emotion regulation [66]. Since basic emotions are mostly related to negative affect, the impaired experience of complex emotions in individuals with PD implicates processing positivity, with a potential impact on patient's subjective and psychological well-being and their interpersonal

relationships [67]. On the other hand, recent evidence suggests a relationship between the emotional status of PD patients and their performance on gait, with an unfavorable impact of deficits in positive feelings on several aspects of locomotion, namely walking speed, arm swing, flexion of posture, step length and gait initiation reaction time [68].

## Non-motor correlates of subjective emotional experience and PD phenotypes

Among the non-motor domains evaluated in the present study, strong correlations were found between subjective emotional experience and the autonomic and cognitive dimensions. Urinary symptoms were the most common autonomic complaints of PD patients, and their prevalence was associated with less intense and less pleasant feelings of Amusement. The most frequent urinary symptoms in PD are suggested to be associated with the loss of dopaminergic neurons in the nigrostriatal pathway [69], overlapping with limbic neurocircuitry engaged in reward processing. On the other hand, a decline in global cognition and greater involvement of the GI tract were strongly associated with a reduced occurrence of Tenderness. Consistent with this, degeneration of the BF, implicated in affiliative feelings, and higher frequency of GI symptoms, have been associated with worse cognitive performance and faster progression to dementia in PD [60,70]. Besides, gut microbiome-derived neurotoxins have been found in the hippocampus and superior temporal lobe neocortex of Alzheimer's disease brains, providing evidence for a link between GI impairment and disruption of nodes from neurocognitive networks that overlap with neural circuitry of affiliative emotions [59,71,72]. Meanwhile, an altered hedonic valuation of tender and disgusting experiences in male patients with PD having higher rates of sexual symptoms may involve the lateral hypothalamic orexin pathway, implicated in emotional valence processing and male sexual behavior [69,73,74].

Data-driven clustering approaches have been applied to identify the co-occurrence of clinical characteristics within groups of PD patients, with results pointing to the existence of two to five clusters that suggest distinct disease subtypes [75]. Our findings are in line with previous studies reporting non-motor phenotypes in PD with a dominant expression of urinary symptoms [76,77], thermoregulatory dysfunction and/or depression [77–79], and coexisting RBD with amnestic impairment [80], possibly indicating widespread disruption of brain networks organized across the "brainstem" (cluster 1), "limbic" (clusters 2 and 3), and "cortical" (cluster 4) levels, in agreement with similar pathophysiological subtypes proposed for PD [81,82]. Specific patterns of overlap between subtypes were also revealed by the clusters, however. The profile represented by cluster 3, with prominent deficits in the subjective feeling of emotion, may be a novel finding of the present work. Our results support the hypothesis that urinary symptoms might be a clinical marker of a more severe PD phenotype [76], as patients in *cluster 1*, who had the oldest age of disease onset, exhibited the highest prevalence of urinary dysfunction as well as the greatest global autonomic disturbances, in addition to the highest overall non-motor burden, all of which may reflect greater striatal degeneration [83], but also significant deficits in multiple neurotransmitter systems, including noradrenaline deficiency resulting from selective degeneration of neurons of the locus coeruleus and sympathetic ganglia [84].

Our results also support previous findings suggesting a heterogeneous pattern of autonomic disturbances among PD individuals, with prevalent cardiovascular and gastrointestinal symptoms in the cognitively impaired patients of cluster 1, in contrast to a more significant relative contribution of thermoregulatory and pupillomotor dysfunction in the members of cluster 2 [32]. The limited capability of the SCOPA-AUT questionnaire in capturing pupillary and thermoregulatory abnormalities, may have led to an underestimation of the actual degree of

impairment in these autonomic domains; therefore, they did not reach the level of statistical significance [32]. Besides significant urinary complaints, patients in *cluster 2* also reported the highest rate of depression and had the youngest age at disease onset, the most prolonged disease duration, and were taking the highest dose of antiparkinsonian medication, in complete agreement with previous reports except that these individuals were all cognitively intact [78]. However, this is in line with findings that cognitive impairment is less frequent in early-onset PD [85]. Strong associations between autonomic and depression symptoms have been confirmed in PD [86], with higher levels of depression in individuals affected by thermoregulatory and pupillomotor disorders [79,87]. Moreover, improvement of autonomic regulation in PD patients after six months of subthalamic nucleus-deep brain stimulation, which resulted in being significant only for the urinary and thermoregulatory functions, was also found to be associated with improvement in a depressive mood, further supporting an overlapping physiology between autonomic and depression symptoms [88]. Coexisting depression and selectively impaired feeling of Amusement align with the finding that dysphoria, a state of generalized unhappiness, is a marker of widespread affective dysfunction in PD patients, reflecting high comorbidity between psychological symptoms [89]. These manifestations suggest the implication of networks integrating the *hypothalamus* [90], a subcortical component strongly related to the mesolimbic dopaminergic reward system that plays a crucial role in humour processing [56], depression [91], thermoregulation [92], and male sexual behavior [69] (a high score was reported by the male member assessed for sexual symptoms). Peripheral mechanisms, which may be associated with the length of levodopa exposure, involving the cholinergic sympathetic, cholinergic parasympathetic, and noradrenergic sympathetic branches, may also contribute to the autonomic alterations revealed in this cluster [93].

Our findings support as well distinct patterns of mental deterioration among the study patients, compatible with two different cognitive syndromes described in PD [94], with significant impairment in the domains of attention/working memory and executive function in cluster 1, in contrast to memory and visuospatial skills in cluster 4. All patients in *cluster 4* were male participants who exhibited probable RBD, amnestic cognitive impairment, and a more significant relative contribution of gastrointestinal symptoms, all in keeping with non-motor features reported to be prevalent in PD presentations dominated by cholinergic dysfunction [95]. In support of our results, strong associations between the presence of RBD and cognitive impairment, including the domains of episodic verbal memory and visuospatial abilities, have been reported in non-demented individuals with PD, with men being at higher risk of poorer cognitive performance [80]. Consistent with the "cholinergic phenotype", emotional deficits in this cluster were selectively associated with the experience of Tenderness, which plays an affiliation function presumably mediated by cholinergic neurotransmission [51]. Of note, affiliative emotions evoke similar neural responses beyond the hedonic value of the affective experience [59]. In line with this, different subtypes of Sadness have been proposed in terms of their eliciting situation, *i.e.*, loss of someone (affiliative) *versus* failure to achieve a goal (non-affiliative), confirming that the two of them produce distinct subjective and physiological responses [96]. To account for the dissociation of Sadness reactivity across affiliative and non-affiliative experiences, we further evaluate the occurrence of the "affiliative emotions". In this category, we included Tenderness and Sadness, which have been found not to be easily distinguishable in some affective contexts [97]. The selectively impaired feeling of these emotions could possibly reflect a reduced ability of PD patients to engage in affective experiences that play a role in attachment behavior (*i.e.*, mixed emotional states associated with empathic feelings), as the film clips intended to elicit Tenderness or Sadness contained scenes featuring affiliative content (*e.g.*, affection, sexual attraction, admiration, compassion). In particular, both film excerpts targeting Sadness recreated scenarios of personal or social loss. Overall, the non-

motor manifestations in this cluster suggest a predominant dysfunction of the cholinergic *basal forebrain* complex and its connected cortex, implicated in tender and sad affiliative feelings [59,98], sleep regulation [99], and, together with the hippocampus, in episodic memory formation [100]. In particular, successful memory performance has recently been shown to rely on strong functional connectivity between the BF and insular cortex, a core area of socioemotional processing [101]. Our findings are also consistent with those of previous studies in PD showing that left-side motor symptom onset is associated with greater deficits in verbal memory, visuospatial analysis, and poorer sleep quality [102,103], regarded as the most important predictor of RBD in these individuals [104].

## A dominant emotion phenotype in PD?

Subjective emotional experience, the feeling, is the essence of an emotion [105]. Patients in *cluster 3* reported the highest impairment in subjective emotional feeling, displaying modifications in the experience of Amusement, Tenderness, Sadness, Disgust and Fear, despite less involvement of the autonomic, mood, sleep, and cognitive domains. Amusement experience was most impaired in these patients, who generally presented right-sided motor symptom predominance that, most likely yet not unequivocally, suggest contralateral dopaminergic defects [106]. This is in accordance with prior reports from an fMRI study presenting evidence that humor engages a network of dopaminergic regions following a pattern of left-lateralization [56]. In agreement with this, the highest levels of Amusement were reported by the patients in cluster 4, who in fact exhibited a significantly higher prevalence of left-sided asymmetry in motor symptoms (*i.e.*, predominant right-lateralized brain impairment) compared to cluster 3. Disgust unpleasantness was also reduced in cluster 3 compared to cluster 4, where patients experienced the strongest feeling of Disgust, in line with a previous report showing that, after damage to the insula and basal ganglia, right-lesioned male patients showed increased disgust composites, while left-lesioned male patients presented attenuated disgust composites, as compared to each other and controls [107]. In agreement with our findings as well, a previous research showed that PD patients with significantly impaired olfactory function rated stimuli that smelled like rotten eggs as less unpleasant than controls [108], whereas another study found that individuals with PD reported less disgust feelings towards poor hygiene and spoiled food than healthy controls [16]. Consistent with this, we found an inverse correlation between all aspects of Disgust subjective experience (occurrence, intensity and unpleasantness) and LEDD, possibly reflecting a link between reduced feelings of revulsion and greater olfactory dysfunction, which has been related to worse disease severity and, consequently, higher needs of antiparkinsonian medication [109].

Comparable to the lower occurrence of Sadness in clusters 3 and 4, the same cohort of male patients with damage to the insula and basal ganglia analyzed in the aforementioned study, also exhibited a significantly reduced experience of Sadness, but not Fear [107]. In keeping with previous research on subjective experience and recognition of emotion [16,17], patients in cluster 3 experienced the highest intensity levels of Fear, which has been suggested to result from the overdose of less compromised circuits (*e.g.*, mesolimbic projections to the amygdala) with the administration of the levodopa dosage required to alleviate the motor symptoms [8,17]. Another possibility is that PD patients with olfactory deficits may compensate their inability to detect environmental hazards through olfaction by an increased ability to detect Fear, as reported in patients with congenital and acquired anosmia [110]. The PD patients in one of the mentioned studies also displayed elevated trait anxiety, that was associated with more intense Fear experience [16]. In our study, male patients who felt higher levels of Fear also reported less occurrence of Tenderness, an important component of empathy [111]. In a

recent experiment, exposing participants to fearful imagery was sufficient to reduce their empathic feelings for others, suggesting that Fear might play a role in attenuating empathy [112]. Since reduced empathy has been reported in PD [63], an open question is to what extent the increased levels of Fear in patients with PD may contribute in shaping their experience of Tenderness and other empathic emotions. Taken together, the significant deficits in emotional feeling revealed in cluster 3 suggest a widespread involvement of the dopaminergic and cholinergic mesolimbic-BF circuitry [56,59,113], in addition to the insular cortex, a key region involved in the pathogenesis of non-motor symptoms in PD [114], associated with olfaction [115], aversion processing [116], affiliative feelings (especially with negative valence) [59], empathy [117], memory [101], and visual perception [118].

Despite very strong agreement of the study outcomes with the existing literature, some discrepancies and their possible implications warrant further consideration. Cognitive impairment with amnestic symptoms, presumably reflecting a posterior cortically based deteriorating process [119], was common in all patients of cluster 4, even though their average age was younger than that reported for cognitively compromised PD patients at greater risk of dementia (age $\geq$ 72 years) [94]. Consistently, these individuals did not display poor semantic verbal fluency (difficulty naming animals), another key factor predicting subsequent worsening of mental health compatible with the "posterior cortical" syndrome [94]. However, these patients did exhibit changes in the domain of memory, in keeping with prior work highlighting the relevance of memory complaints in predicting the development of dementia in PD [120,121]. Still, since PD dementia is characterized by the addition of cortical dysfunction upon fronto-subcortical deficits [122], semantic knowledge in these patients may become affected with further progression of the disease. Cholinergic BF degeneration is an important contributor to cognitive deterioration in PD. In particular, baseline atrophy and longitudinal changes in the nucleus basalis of Meynert (Ch4) have been associated with worse global cognition and more specifically, with attention and visuospatial impairment [60,123]. Furthermore, volume loss in this posterior part of the BF has been found to be followed by longitudinal atrophy in the anterior regions [60]. Consistent with this, patients classified as having smaller Ch4 volumes showed more sever and rapid decline in recall memory and semantic fluency [124]. Comparable to the associations reported for Ch4 structural changes, we found that Tenderness occurrence correlated significantly with attention and visuospatial functions. Moreover, in combination with delayed verbal recall and semantic fluency, the ability to experience tender feelings explained 83% of the variance in global cognition among patients without prominent frontal damage. We found Tenderness experience and recall memory to have a similar though independent contribution to global cognition, in line with prior work highlighting that disturbances in memory and emotion are independently related to cognitive impairment [125]. The interrelation we report here may reflect the discrete contribution of distinct functional networks, which subserve affiliative behavior, memory, and semantic cognition, to overall BF neuromodulatory dynamics [59,72,126,127]. A main implication of our findings is that, a marked and *selective inability to experience Tenderness* might be one of the earliest predictors of worse cognitive outcomes in PD. In fact, differences in the prevalence of sentiments that may promote affiliative interactions (*e.g.*, Tenderness and Sadness), in contrast to deficits in semantic memory and delayed recall, made possible a better differentiation of patients in cluster 4. Consistent with our results, diminished subjective experience of Sadness has been recently described in patients with frontotemporal dementia, especially in the aphasic variant, distinguished by a loss of semantic knowledge, and in the behavioral variant, characterized by emotional blunting [128]. This latter variant also displayed impaired feeling of affiliative emotions, which has been linked to frontopolar and septal damage [129]. Longitudinal studies are

needed to unveil the extent to what selective deficits in affiliative feelings might represent early biomarkers of a prodromal PD dementing process.

We acknowledge some limitations. The study patients were under regular medication, which may lead to underestimating the deficits in emotional processing. However, no significant effect of dopaminergic treatment on emotional reactivity has been previously reported in PD [17,49,130]. Considering that some patients were taking antidepressant drugs, depression prevalence may have been underestimated, which may have hidden or weakened potential associations between depression scores and other features. Data were insufficient to assess women's sexual SCOPA-AUT subscale. Clinical evaluation was performed only by self-report scales. A wearing-off questionnaire (*e.g.*, WOQ-19) would be useful to administer in future studies to account for possible fluctuations of non-motor symptoms associated with chronic levodopa therapy in PD [131]. Since our study relied on participants' cooperation to engage with the content of the films, it would have been convenient to assess their degree of engagement during the experiments. A possibility is that a larger experimental group is necessary to increase the chances of detecting a significant effect of PD on hedonic tone, as valence ratings in response to overall pleasurable stimuli resulted in being slightly lower in the PD group ($p < 0.06$). Moreover, it would have been advantageous to measure emotion intensity and valence using rating scales of similar size to facilitate their interpretation. Further research on a larger sample size, including the analysis of fMRI data and quantitative assessment of non-motor features and their correlation with motor symptoms, is needed to confirm and extend our findings.

## Conclusions

Feelings matter in PD. We found the subjective experience of complex emotions to be impaired in the PD individuals. The inventory of emotions produced in the patients was restricted by a lower occurrence of Tenderness, whereas their emotional experience was characterized by less intense feelings of Tenderness and Amusement. Impaired experience of Tenderness (occurrence) and Amusement (intensity) was found to be associated with autonomic dysfunction, namely, the prevalence of gastrointestinal and urinary symptoms, respectively. Besides, the occurrence of Tenderness was related to the overall cognitive status of the patients. The results of this study support a substantial diversity among the clinical profiles that arise from the co-occurrence of non-motor characteristics, highlighting the relevance of emotional disturbances in delineating PD subtypes. Our findings further suggest the possible existence of a PD phenotype with greater modification of subjective emotional experience, in particular low intensity of Amusement, low occurrence of both Tenderness and Sadness, and probably other changes that have been previously described in PD, such as increased intensity of Fear and reduced unpleasantness of Disgust. The present work provides additional evidence for the non-motor heterogeneity of PD, with potential clinical implications for the achievement of precision medicine. Future studies with larger sample sizes are needed to explore other relevant clinical manifestations and their association with a broader set of positive and negative affective states, as well as basic and complex emotions. In addition to self-report ratings of emotional experience, studies analyzing behavioral features and physiological measures of the peripheral autonomic nervous system are required to enhance our understanding of emotion in PD.

## Acknowledgments

We would like to thank all participants for their valuable contribution to this study, especially the PD patients for their unconditional support and courage. We are grateful to Dra. Cristina

Fernández Megías for facilitating the battery of film excerpts used in the study and for her advice on how to use it, as well as Prof. Carmelo Vázquez Valverde for facilitating the Spanish version of the BDI-II. Authors would also like to thank CNEURO, especially Elsa Santos Febles and Tania Y. Aznielle Rodríguez, at the Department of Software Development, for their valued help in the organization of the research, as well as the Department of Electronics, Department of Design and Technology, Department of Cognitive Neurosciences, Department of Radiology and Brain Mapping, and the General Maintenance Brigade for the logistical support. We gratefully acknowledge the clinicians from the Neurodegenerative Diseases and Movement Disorders Clinic at CIREN for their support in the recruitment of patients. The authors of the study greatly appreciate the thorough review and valuable recommendations from the reviewers, which contributed to significant improvements of this paper.

## Author Contributions

**Conceptualization:** Claudia Carricarte Naranjo, Andrés Machado, Hichem Sahli, María Antonieta Bobes.

**Data curation:** Claudia Carricarte Naranjo, Claudia Sánchez Luaces.

**Formal analysis:** Claudia Carricarte Naranjo.

**Funding acquisition:** Hichem Sahli.

**Investigation:** Claudia Carricarte Naranjo, Claudia Sánchez Luaces, Ivonne Pedroso Ibáñez.

**Project administration:** Claudia Carricarte Naranjo, Claudia Sánchez Luaces, Hichem Sahli.

**Resources:** Ivonne Pedroso Ibáñez, Andrés Machado, Hichem Sahli, María Antonieta Bobes.

**Software:** Claudia Carricarte Naranjo.

**Supervision:** Ivonne Pedroso Ibáñez, Andrés Machado, Hichem Sahli, María Antonieta Bobes.

**Visualization:** Claudia Carricarte Naranjo, María Antonieta Bobes.

**Writing – original draft:** Claudia Carricarte Naranjo.

**Writing – review & editing:** Claudia Sánchez Luaces, Ivonne Pedroso Ibáñez, Andrés Machado, Hichem Sahli, María Antonieta Bobes.

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
