## [Decision Letter · Decision Letter 0]

30 Aug 2022

PONE-D-22-11791Beyond shallow feelings of complex affect: non-motor correlates of subjective emotional experience in Parkinson's diseasePLOS ONE

Dear Dr. Carricarte Naranjo,

Thank you for submitting your manuscript to PLOS ONE. After careful consideration, we feel that it has merit but does not fully meet PLOS ONE’s publication criteria as it currently stands. Therefore, we invite you to submit a revised version of the manuscript that addresses the points raised during the review process.

We look forward to receiving your revised manuscript.

Kind regards,

Vincenzo De Luca

Academic Editor

PLOS ONE

Journal Requirements:

Reviewers' comments:

Reviewer's Responses to Questions

**Comments to the Author**

1. Is the manuscript technically sound, and do the data support the conclusions?

Reviewer #1: Partly

Reviewer #2: Partly

2. Has the statistical analysis been performed appropriately and rigorously? 

Reviewer #1: I Don't Know

Reviewer #2: I Don't Know

3. Have the authors made all data underlying the findings in their manuscript fully available?

Reviewer #1: Yes

Reviewer #2: Yes

4. Is the manuscript presented in an intelligible fashion and written in standard English?

Reviewer #1: Yes

Reviewer #2: Yes

5. Review Comments to the Author

Reviewer #1: Beyond shallow feelings of complex affect: non-motor correlates of subjective emotional experience in Parkinson’s disease

This interesting study examines potential emotion-related abnormalities in Parkinson’s. The paper is extremely well-written and concise. It provides a clear rationale for the work. Whereas motor phenotypes have been fairly well-characterized in Parkinson’s, less is known about affective and autonomic dimensions of the disorder, though there is clearly evidence of costly emotional changes associated with the disease. Characterizing these emotional problems and potential subtypes may help inspire more targeted disease-modifying therapies.

1)I always appreciate a concise introduction, but here I think this one could include a section on what is currently been done, and what the gaps are with respect to emotion in PDThere are studies on blunted reactivity (Bowers et al., 2006, Brain), reviews on emotion and PD (Sotjiu & Rusconi 2013 Front. Psychol; Peron et al., 2011 Movement disorders), emotion processing across different levels (Enrici et al., 2015 Plos one), reactivity to facial emotion (Ruzicka et al., 2019 Eur Neuropsychopharm), emotion detection and empathy (Martinez et al., 2018, Front Aging Neurosci). These are just a few.

2)Related to controlling for other factors:

a.There is virtually no information on the control group. Were they matched in terms of age, sex, and IQ? Did they do a BDI? Were they taking any psychotropic meds? What was the age of the PD group? A table would be helpful in this regard.

b.Participants were taking their usual dose of meds. Did they quantify this and look at associations with emotion? There are some systems that might be more sensitive to dopaminergic manipulations than others. It seems relevant.

c.On a related note, for some of the analyses, the authors controlled for the effect of disease duration. Did they take into account the severity of motor symptoms as well whilst on and off the medication? It sounds like they would have this data and it would be useful to examine the extent to which they covary with the emotional symptoms. If there is a dissociation, that would be of interest since it would argue for a subtype rather than an effect of disease severity.

d.The issue concerning the potential effects of psychotropic medication is not yet adequately addressed. The medications taken by 10 of the patients are known to affect emotional reactivity. The authors note (line 149-150) that patients were 12 hours from their last intake; however, the half-life of some of these drugs is much longer (e.g., clonazepam 30-40 hours; olanzapine around 30 hours). The authors report the half-life of clonazepam as 8-12 hours (lines 390), but I think this is an underestimate (e.g., see DeVane et al., 1991, Psychopharm Bull; Crevoisier et al., 2003 Eur Neurol). It would be critical to present some data concerning whether those taking psychotropic meds showed the same pattern as those that did not.

e.The results, conclusions and discussion should be revised in accordance with the above to ensure that they are in agreement with the data.

3)It is unfortunate that the intensity of the experienced emotion was rated on a 7-point scale, but the emotional valence was rated on a 9-point scale. To avoid confusion, making these the same would have been advantageous.

4)The introduction mentions autonomic arousal. In addition, physiological data were also collected during the experiment, but for some reason, they were not reported here. Given these data seem relevant (and provide additional information on emotional reactivity), it would seem they should also be reported here. The authors should provide a clear rationale for not presenting the data. I must admit, I have trouble thinking of a valid reason for omitting it.

5)I do not understand how the a set of features were identified for cluster analysis based on the description provided in lines 124-129. I was not sure whether there is a line missing or perhaps just more details are needed. A couple of lines here on what k means cluster analysis is and what it does would be helpful to readers as would a rationale for why it was applied with this data.

6)I would suggest some revisions to the organization of the results to make them more clear and the number of tests transparent. For example:

a.The presentation of results from the ANOVA conducted on the intensity ratings were hard to follow (lines 170-184). It would be simplest to describe what the ANOVA was exactly (e.g., was it a 2(Group: PD, control) X 2(Participant Sex) X 7(film emotion) first. Then describe each of the main effects Group, then sex, then film emotion. Then describe all interactions therein. Only those effects involving a significant interaction should then be unpacked with follow-up tests.

b.Lines 196-210: A number of correlations are presented. These should be presented in a table. It should also be made clear the number of correlational tests performed, the number of tests that were input for Benjamini Hochberg correction, and which correlations survive correction.

i.I would suggest focusing on correlations between the experimental measures and the symptom measures. You could present correlations between clinical measures (e.g., MoCA and SCOPA-AUT and its subscales), but this is a separate issue and probably belongs in a different section. What is of greatest interest to the reader is the relationship between these symptoms and the experimental measures of emotional experience.

7)On line 259: there is suggestion that they found “no impairment in the subjective experience of basic emotions”. Could the authors clarify which metric determined this? If it is valence rating, it is important to consider effect sizes as I suspect that the issue might be one of power than that an effect is absence. One could do Bayesian approaches to assess the likelihood that the null is true if the authors want to make strong conclusions in this regard.

Reviewer #2: Dear Editors and Authors,

Thank you for the privilege of reviewing the manuscript titled “Beyond shallow feelings of complex affect: non-motor correlates of subjective emotional experience in Parkinson's disease”. In it, the authors seek to evaluate the affective changes in Parkinson’s disease (PD) by examining subjectively reported emotional responses to movie clips. They also seek to determine the relationship between non-motor/autonomic symptoms of PD, and these emotional responses.

I generally thought this was an interesting, and well written paper, and would support its publication so long as some of the issues below can be addressed.

In particular, the issue regarding the possible impact of the long-lasting benzodiazepines used by many of the PD patients in this study needs to be further discussed or analyzed, given that these medications can possibly affect emotional response, and that a study which found different results that this paper (citation 9) did not have any patients on these medications. As well, I will ask the Editor to ensure that either another reviewer, or external statistical expert, is able to determine the validity of the statistical analysis used, as they are beyond my expertise.

Strengths

1.I found the paper generally very well written and easy to follow.

2.I think this is an interesting and helpful topic to further understand, and a good application of the methodologies used

3.The authors do a good job discussing their results and interpreting it with respect to prior literature.

4.Studying a non-native English speaking population helps add needed diversity to the literature.

Major Issues

1.My computational expertise are more in machine learning methodology than in statistics. While I generally thought the statistical analysis was done correctly, I will ask the editor to ensure either an outside expert, or another reviewer with such expertise, is able to confirm the validity of the statistical methods used, especially given the complexity and relatively small sample size.

2.K-means requires that users specify the number of clusters they wish to apply the algorithm to. From what I can tell, the authors choose 4 clusters given prior work that used this number (citation #20). Looking at this reference, it seems there was some similarity in the data. I’d ask that the authors please further validate why using this number is okay – is that prior work’s dataset/population so similar that this number can be used again? Have other work also used this number? Otherwise, it might be preferable to do some analysis yourselves that this is an optimal number of clusters (as was done in citation 20).

3.A larger concern I have regarding the validity of the results lies in the impact of the subject’s pharmacology. 10 patients are taking psychotropic medication. However, the authors mention that “Excluding the four patients under antidepressant and antipsychotic treatment from the analysis did not alter these results confirming no significant effect of the psychotropic drugs on the present findings” and in the limitations note “however, since [clonazepam’s] effects last about 8 to 12 hours, no important influences on the study measures were expected”. My understanding and experience as a psychiatrist is that while the acute anxiolytic/sedative effects might be though to last 8 to 12 hours, they do not stop having any psychotropic effect after this period. As well, did subjects stop taking their medications more than 12 hours before the testing? Clonazepam has a half-life 19-60 hours, so at steady state is still going to be at high serum levels even 12 hours last dose. It is sometimes used in anxiety and epilepsy disorders being prescribed once daily. The authors cite that it is not known to have an effect on REM sleep parameters. However, my understanding is it can affect cognition, especially in the elderly, as well as emotions. I believe the authors must better establish that clonazepam/alprazolam use is not contributing to the differences found in the study. This is especially important given the relatively large number of participants on them, and given that the differences found were in more complex emotions, which could be affected by the cognitive-effects of these long-lasting benzodiazepines in this elderly population. Analysis of PD subjects without those on any of the psychotropic medications (or not on a benzo) would be helpful, or further citation of prior work showing that these medications would not be expected to affect the complex emotional responses like tenderness. I note that in citation 9, where they found that the emotion response in PD was unimpaired, none of the subjects seem to have been on benzodiazepines, though some are on antidepressants; citation 11 does not seem to discuss this. Citations 10 and 20 seemed to exclude patients on psychotropic medication or with psychiatric history I think, though this may not be naturalistic.

4.Somewhat related to the above, I’d ask that the authors further address the possible impact of depression on the results, especially if the patients had a long, lifetime history of depression which preceded the PD. Could depression not attributable to PD be explaining some of the results found by the authors? Prior literature, or analysis of those without depression, could be helpful to ensure that PD is the cause of the findings. I do recognize that PD can cause depression, of course, but I think some more consideration would be helpful to help tease this apart, though I acknowledge the authors did do this to some extent by excluding those on antidepressants and finding the same results.

Minor Issues

1.“The sex of participants had not a significant effect on the differences observed in the mean emotion intensity reported for these target emotions” I would suggest “The sex of participants did not have a …” as the current wording reads a bit odd to me.

2.Table 2 – for the drugs, there are percentage signs in the reported numbers, but not for other percentages reported.

6. PLOS authors have the option to publish the peer review history of their article (what does this mean?). If published, this will include your full peer review and any attached files.

Reviewer #1: No

Reviewer #2: **Yes: **John-Jose Nunez

---

## [Author Response · Author response to Decision Letter 0]

24 Jan 2023

We have uploaded a file with a line-by-line response to all the Reviewers' concerns. Please see the "Response to Reviewers" document.

---

## [Decision Letter · Decision Letter 1]

5 Feb 2023

Beyond shallow feelings of complex affect: non-motor correlates of subjective emotional experience in Parkinson's disease

PONE-D-22-11791R1

Dear Dr. Carricarte Naranjo,

We’re pleased to inform you that your manuscript has been judged scientifically suitable for publication and will be formally accepted for publication once it meets all outstanding technical requirements.

Kind regards,

Vincenzo De Luca

Academic Editor

PLOS ONE

Additional Editor Comments (optional):

Reviewers' comments:

Reviewer's Responses to Questions

**Comments to the Author**

1. If the authors have adequately addressed your comments raised in a previous round of review and you feel that this manuscript is now acceptable for publication, you may indicate that here to bypass the “Comments to the Author” section, enter your conflict of interest statement in the “Confidential to Editor” section, and submit your "Accept" recommendation.

Reviewer #2: All comments have been addressed

2. Is the manuscript technically sound, and do the data support the conclusions?

Reviewer #2: Yes

3. Has the statistical analysis been performed appropriately and rigorously? 

Reviewer #2: Yes

4. Have the authors made all data underlying the findings in their manuscript fully available?

Reviewer #2: Yes

5. Is the manuscript presented in an intelligible fashion and written in standard English?

Reviewer #2: Yes

6. Review Comments to the Author

Reviewer #2: I believe the authors did a great job addressing feedback from both reviewers, and am happy to support publication.

7. PLOS authors have the option to publish the peer review history of their article (what does this mean?). If published, this will include your full peer review and any attached files.

Reviewer #2: **Yes: **John-Jose Nunez

---

## [Editor Report · Acceptance letter]

15 Feb 2023

PONE-D-22-11791R1 

Beyond shallow feelings of complex affect: non-motor correlates of subjective emotional experience in Parkinson's disease 

Dear Dr. Carricarte Naranjo:

I'm pleased to inform you that your manuscript has been deemed suitable for publication in PLOS ONE. Congratulations! Your manuscript is now with our production department. 

Kind regards, 

on behalf of

Dr. Vincenzo De Luca 

Academic Editor

PLOS ONE